# The Effects of Hybrid Tele Airway Clearance in Bronchiectasis Patients: A Case Series

**DOI:** 10.3390/reports7030057

**Published:** 2024-07-19

**Authors:** Aung Aung Nwe, Nimit Kosura, Chatchai Phimphasak, Pornthip Barnludech, Si Thu Aung, Worawat Chumpangern, Chulee Ubolsakka-Jones

**Affiliations:** 1School of Physical Therapy, Faculty of Associated Medical Sciences, Khon Kaen University, Khon Kaen 40002, Thailand; aungaung.n@kkumail.com (A.A.N.); joneschulee@gmail.com (C.U.-J.); 2Innovation to Improve Cardiopulmonary and Physical Performance (IICP) Research Group, Khon Kaen University, Khon Kaen 40002, Thailand; knimit@medicine.psu.ac.th (N.K.); pornthip_barnludech@kkumail.com (P.B.); worawat@kku.ac.th (W.C.); 3Department of Physical Therapy, Faculty of Medicine, Prince of Songkla University, Songkla 90110, Thailand; 4Unit of Physical Therapy, Department of Rehabilitation, Chulaborn Hospital, Bangkok 10210, Thailand; 5Department of Tropical Medicine, Faculty of Medicine, Khon Kaen University, Khon Kaen 40002, Thailand; sithuaung@kkumail.com; 6Pulmonary and Critical Care Medicine, Department of Medicine, Faculty of Medicine, Khon Kaen University, Khon Kaen 40002, Thailand

**Keywords:** physical therapy, tele, airway clearance, bronchiectasis

## Abstract

This study aims to evaluate the preliminary effects of a hybrid tele-supervised airway clearance protocol on secretion clearance, health-related quality of life, and patient satisfaction in bronchiectasis patients. A single-arm experimental pilot case series with three participants was conducted, involving six ACT sessions over three days, including one onsite supervised session and five tele-supervised sessions. Assessment measures comprised sputum expectoration, COPD assessment test (CAT), and participant satisfaction ratings. The results showed increased sputum expectoration rates during each ACT session, alongside notable improvements in CAT scores (reductions of 16, 8, and 8 points for each participant). Participants expressed high satisfaction with tele-supervised sessions and reported increased confidence in independent ACT performance post-program. The findings suggest that the hybrid ACT program may be a promising avenue for enhancing bronchiectasis management. However, further research with larger sample sizes and rigorous control groups is necessary to validate its efficacy and broader applicability.

## 1. Introduction

Bronchiectasis is a chronic lung condition characterized by the permanent dilation of the bronchi, leading to symptoms such as coughing, sputum production, and recurrent chest infections [1]. Multimodal management has been recommended for this disease, including long-term antibiotic treatment, anti-inflammatories, mucoactive drugs, bronchodilators, surgical options, and respiratory physiotherapy [2].

In Thailand, patients with bronchiectasis are typically referred to respiratory physiotherapy by physicians. Healthcare delivery for these patients usually occurs in two settings. The first is a formal referral to the physiotherapy department. The second is a one-stop service, where patients receive comprehensive care, including medical, pharmacy, nursing, dietary, and physiotherapy management, all within a single visit typically lasting around 2 to 3 h. Although the one-stop service has many advantages, the time limitations can affect physiotherapy sessions.

Airway clearance techniques (ACTs) are fundamental to respiratory physiotherapy and crucial for enhancing secretion clearance from the lungs, especially in patients with chronic productive cough, such as those with bronchiectasis. Popular ACTs include manual chest physiotherapy, forced expiratory techniques, and oscillatory positive expiratory pressure devices [3]. However, access to these interventions can be limited, particularly in remote or resource-constrained settings, leading to challenges in delivering consistent and timely care to patients. 

Based on our experience, ACTs require skilled training, and a single supervised session is insufficient for patients to develop the necessary skills and confidence to perform these techniques correctly at home over the long term. Additional sessions of supervision in performing ACTs might be beneficial.

Telemedicine provides a medical approach at home, replacing the traditional face-to-face approach of medicine. Telemedicine makes healthcare services more accessible, particularly for individuals facing barriers such as geographical distance, travel constraints, or social isolation. Its rapid expansion occurred during the COVID-19 pandemic [4]. In the context of chronic lung disease, telerehabilitation has become popular. Recent studies have shown that pulmonary rehabilitation mainly utilizing physical exercise via telemedicine is safe and can be considered as an option to improve exercise capacity and quality of life and reduce anxiety and depression in patients with COPD [5]. 

ACTs are primarily aerosol-spreading procedures, necessitating more stringent isolation measures [6]. In a one-stop service setting, the limitations of isolation facilities make it challenging to perform ACTs. Additionally, it is not suitable to remove masks while teaching ACTs, making it difficult to assess correct performance. Tele-supervised ACTs might be more suitable as they reduce the risk of aerosol spreading and lower the risk of infection. Moreover, the implementation of tele-supervised airway clearance, delivering airway clearance techniques, remains limited. 

For the reasons outlined above, tele-ACT, which delivers ACT services to patients with bronchiectasis, may be beneficial. Therefore, we aimed to report the preliminary efficacy of hybrid tele-supervised airway clearance in patients with bronchiectasis in terms of secretion clearance, health-related quality of life, confidence, and satisfaction.

## 2. Detailed Case Description

This study was a single-arm experimental pilot case study involving the development and delivery of physiotherapist-supervised airway clearance to patients with bronchiectasis, who typically have sputum production greater than 10 mL/day. The intervention was delivered via tele-video calls and included three participants. This study was carried out between 8 November 22 and 10 September 2022 in Thailand. The ethical approval of this study was obtained from Center of Ethics in Human Research, Khon Kaen University, Thailand (HE651328). Informed consent was obtained from the participants before the start of the study.

### 2.1. Subjects

The recruitment of the participants was performed at Srinagarind Hospital, Thailand. The first three individuals who met the criteria and had an interest in participating in hybrid tele airway clearance are included in this pilot case report. The inclusion and exclusion criteria are shown in Table 1.

### 2.2. Procedure

Preparation Phase: On the day of recruitment at the chest clinic, several procedures were carried out. Firstly, a detailed explanation was provided regarding the hybrid tele airway clearance program. Secondly, demographic data were collected from participants, and sputum containers were provided for the run-in phase. Additionally, participants were instructed to suspend their mucolytic drugs for the duration of the study, while other medications were to be continued as usual. To regulate and prevent airway hydration, participants were advised to adhere to a minimal daily water intake, as suggested by Gaspar (2011) [7]. Furthermore, discussions were held with participants regarding their preferred method of tele-video communication with researchers, such as Messenger or the Line app. Lastly, participants’ addresses were recorded on Google Maps during this phase.

Run-in Phase: The run-in phase lasted 3 days. During this period, no ACT was carried out. Participants collected their natural expectorated sputum three times per day: from 6 to 9 a.m., 9 a.m. to 6 p.m., and 6 to 9 p.m.

ACT Phase: The pilot airway clearance sessions were conducted twice a day, in the morning and evening, at times convenient for the participants, for three consecutive days. Researchers visited participants’ homes and provided sputum containers for this phase to facilitate sputum collection. Hence, the initial airway clearance session, on the first day, was supervised at the participant’s home. Additionally, role-playing and testing of video calls, as well as adjusting the positioning of the phone and participants for optimal visibility during the video call, were conducted and recommended. Subsequent airway clearance sessions, starting from the second session on the first day, were conducted via video call. Therefore, a total of 6 airway clearance sessions were included in this hybrid tele-airway clearance program.

The airway clearance technique used in this study was the modified forced expiratory technique (mFET). A schematic comparison of the original FET [8] and mFET is shown in Figure 1. The components of mFET in this study included low-lung-volume huffs, mid-lung-volume huffs, high-lung-volume huffs, and breathing control interspersed between huffs.

For the low-lung-volume huff, participants were instructed to take a slow, deep breath and then exhale as slowly as possible. Following this, they were instructed to immediately take a small breath below their tidal volume and then perform a huff. For the mid-lung-volume huff, participants were instructed to breathe normally (tidal breathing) and then perform a huff. For the high-lung-volume huff, participants were instructed to take a deep breath, inhaling as much as possible, and then perform a huff. Breathing control was performed at the patient’s own rhythm.

A set of mFETs comprised the following steps: starting with 3 low-lung-volume huffs followed by 3 breathing control exercises, progressing to 3 mid-lung-volume huffs and 3 breathing control exercises and, finally, performing 3 high-lung-volume huffs followed by 3 breathing control exercises (Figure 2). Ten sets were performed with at least one minute of rest between each set. Two sessions per day (morning and evening) for three days were conducted for this study. Each session lasted around 20–40 min.

### 2.3. Assessments

Expectorated sputum: The participants’ collected expectorated sputum during the run-in phase (Table 2) and ACT phase (Table 3) was centrifuged at 4 degrees Celsius and 4500 rpm for 20 min. Following centrifugation, the supernatant saliva was removed, and the weight was measured in grams.

COPD assessment test (CAT): CAT is a multidimensional health-related quality of life questionnaire. This questionnaire consists of eight items, including cough, sputum, dyspnea on exertion, chest tightness, activity limitation, confidence, sleep quality, and energy. The Thai version of the CAT was assessed before and after the ACT period. 

Satisfaction with and perception of tele airway clearance: At the end of the program, participants were surveyed regarding their satisfaction with performing ACTs via tele-supervision using a 5-point Likert scale consisting of options including “Very Satisfied”, “Satisfied”, “Neither Satisfied nor Dissatisfied”, “Dissatisfied”, and “Very Dissatisfied”. Participants were also asked about their perception of tele airway clearance compared to onsite supervision, such as how they felt about the difference between onsite supervision and tele supervision.

## 3. Results

### 3.1. Demographic Characteristics of Participants

A male and two female participants with bronchiectasis, each with a daily sputum expectoration of more than 40 g (based on a 3-day average from 6 a.m. to 9 p.m.), were included in this study. Detailed demographic characteristics of the participants are shown in Table 4.

### 3.2. Participants’ Sputum Expectoration

The detailed sputum expectoration rates per hour for the three participants are shown in Figure 3. All three participants exhibited a greater increase in sputum expectoration rates per hour during the morning ACT session compared to the morning period in the run-in phase. Similarly, there was a greater increase in sputum expectoration rates per hour during the evening ACT session compared to the evening period in the run-in phase for all participants. The post-2 h period of the morning ACT session still showed greater sputum expectoration rates per hour compared to the morning period in the run-in phase for all three participants. However, in the post-2 h period of the evening ACT session, only participant 3 exhibited a greater increase in sputum expectoration rates per hour compared to the evening period in the run-in phase.

### 3.3. Participants’ CAT Score

Table 5 presents the pre- and post-assessment CAT scores, along with the change scores. The findings indicate a decrease in the total CAT score across all three participants, with changes of −16, −8, and −8 for participants 1, 2, and 3, respectively. Furthermore, reductions in sub-items of the CAT score, including cough, phlegm in the chest, breathlessness on exertion, confidence of leaving home, sleep quality, and energy level, were observed when comparing pre- and post-assessment results.

### 3.4. Satisfactory and Perception of Tele Airway Clearance

All three participants reported being “Very Satisfied” with the tele airway clearance supervision sessions. Additionally, all participants rated feeling the “Same Feeling” and that tele airway clearance was more convenient compared to onsite supervision and gave them the confidence to perform the modified FET without supervision.

## 4. Discussion

We administered a hybrid tele airway clearance program, comprising a single session supervised at home and five sessions supervised via video calls over three days, to three bronchiectasis patients with extensive sputum production. Our findings reveal that all three participants experienced enhanced secretion clearance and improvements in their quality of life.

Telemedicine has been increasingly utilized in managing chronic respiratory conditions, with a particular emphasis on tele rehabilitation, especially in patients with chronic obstructive pulmonary disease (COPD). Recent systematic reviews have highlighted the efficacy of tele rehabilitation, demonstrating improvements in functional exercise capacity, quality of life, and reductions in anxiety and depression among individuals with COPD [5]. 

ACTs are some of the mainstay physical therapy treatments in patients with chronic respiratory diseases with secretion problem like bronchiectasis. Moreover, Lee (2023) suggested that the effect of tele airway clearance should be explored [9]. To the authors’ knowledge, only a few studies have been carried out on tele airway clearance. In the study by Alghamdi et al. (2023), participants were instructed in Acapella and ACBT techniques via tele-platform, which included written information, leaflets, and links to online videos in patients with COPD. After three months, improvements in health-related quality of life, measured by total CAT score, were observed: −1.96 (95%CI: −2.51 to 2.19) for Acapella and −1.81 (95%CI: −4.38 to 2.83) for ACBT [10]. In our study, all three participants showed improved total CAT scores, with changes exceeding the MCID (>3) [11] for bronchiectasis (change scores: 16 in participant 1, 8 in participants 2 and 3) after three days of the program. Our approach, supervised via video call, may have contributed to these better outcomes by allowing for real-time reinforcement and correction of techniques during ACT sessions. However, it should not be ignored that the findings of our study and that of Alghamdi et al. (2023) [10] are on different lung diseases, i.e., bronchiectasis and COPD, although both are chronic respiratory diseases. 

We firmly believe that mastering the correct procedure and instilling confidence in patients to perform ACTs is crucial for long-term adherence, especially in device-free methods like mFET. Our ACT training program, comprising six sessions over three days (including one onsite session at home and five video call supervisions), adequately addresses the need for correcting procedures and boosting patients’ confidence. Notably, a similar number of video supervision sessions were employed in a recent study by Hamidfar et al. (2023), wherein they conducted three to five sessions of video conferencing airway clearance training using the device in bronchiectasis patients, followed by 10 weeks of self-administered airway clearance with physical therapist reinforcement, motivation, and technical assistance every ten days via telecare [12]. This suggests that our tele program is sufficient to ensure patients can perform self-administered airway clearance correctly after completing the program.

In our study, we used the mFET among various ACTs because it is device-free and one of the usual physical therapy interventions in bronchiectasis. Since this technique is device-free, its effectiveness is mainly dependent on it being correctly performed by the patients. The efficacy of the mFET relies on the relationship between the location of secretions and the initial lung volume of huffing. A high-lung-volume huff (full breath) is necessary to clear secretions from the central airway, while a huff at a lower lung volume clears secretions from the peripheral airway (based on equal pressure point theory) [8]. In traditional FET, prolonged huffing is utilized, such as huffing from mid-lung volume to low-lung volume to mobilize more peripheral secretions, and from high-lung volume to mid-lung volume to clear more proximal secretions [8]. However, extension of the huff from mid-lung to low-lung volume may cause coughing, which might limit the clearance of secretions from the peripheral airways and may not be feasible for some patients, especially those with excessive sputum. Therefore, in our study, we modified this technique by implementing low-lung-volume huffing, instructing participants to expire as much air out of the lungs as possible, then take a small breath (starting lung volume under FRC), and then huff to clear more peripheral secretions (Figure 1).

The expectorated sputum is a straightforward and direct indicator of the effectiveness of ACTs. Numerous studies on ACTs in bronchiectasis have used sputum expectoration, measured by weight or volume, as an outcome, demonstrating the impact of a single ACT session [13,14,15,16,17,18,19]. However, the nature of sputum production can vary from day to day and within a single day [20]. Therefore, in our study, we conducted ACT sessions twice daily for three consecutive days to ensure a comprehensive assessment of effectiveness. We found that the rate per hour of sputum expectoration while performing ACTs was greater than at other times on ACT-performing days, as well as on non-ACT days (run-in phase) in all three participants. These findings highlight that our hybrid tele-ACT can effectively clear secretions.

Our hybrid tele-ACT approach, utilizing the modified forced expiratory technique (FET), effectively promotes secretion clearance. Our telemedicine platform allowed for personalized guidance and support during each session. Patients received individualized teaching on proper ACTs, real-time feedback on their performance, and the opportunity to ask questions and address concerns. Patient satisfaction is a key measure of how well a telemedicine approach meets patient needs, and it is an important factor for successful healthcare delivery [21]. In our three patients, they reported being very satisfied with our hybrid tele ACT program. Despite the overall positive feedback, we encountered some challenges during the implementation of the program. Technical issues, such as participants’ lack of familiarity with applications for making video calls, and instances where device batteries ran out during video call supervision, were encountered. However, we promptly addressed these challenges by providing guidance on application usage and recommending that participants ensure their devices were adequately charged before scheduled video calls.

As for the implication of our findings, the hybrid tele-supervised ACTs (airway clearance techniques) can be effective in improving secretion clearance in bronchiectasis patients. Additionally, in low–middle-income countries (LMICs), where access to specialized healthcare facilities may be limited, tele-supervised ACTs could represent a viable alternative to in-person sessions, enabling patients to receive consistent care remotely. Furthermore, implementing similar tele-health interventions in LMICs can enhance patient outcomes without requiring significant infrastructure, thus improving the quality of life for patients with chronic respiratory conditions.

We are also aware of the limitations in our study. Firstly, this study is based on a small sample size and consists solely of case reports. As such, the findings may not be generalizable to broader populations. Additionally, our study lacked a follow-up period to assess the long-term effects of the ACT. Furthermore, the absence of a comparison group limits our ability to attribute the observed outcomes solely to the tele-ACT program. A controlled study design with a comparator group receiving standard care would provide more robust evidence of the program’s efficacy. 

Therefore, we recommend that researchers conduct larger randomized controlled trials across diverse geographic locations to include more participants and ensure generalizability. Additionally, future studies should incorporate cost-effectiveness analyses comparing tele-supervised ACTs with traditional in-person care, taking into account the costs of technology and training. Furthermore, pilot programs should be designed to work within the constraints of current healthcare systems in low- and middle-income countries (LMICs), with a focus on scalability and sustainability.

In conclusion, our study underscores the potential of telemedicine in revolutionizing respiratory care delivery, particularly in enhancing airway clearance techniques for bronchiectasis patients. The positive outcomes observed in secretion clearance, quality of life, and patient satisfaction suggest that tele airway clearance could be beneficial for bronchiectasis patients with secretion problems.

## Figures and Tables

**Figure 1 reports-07-00057-f001:**
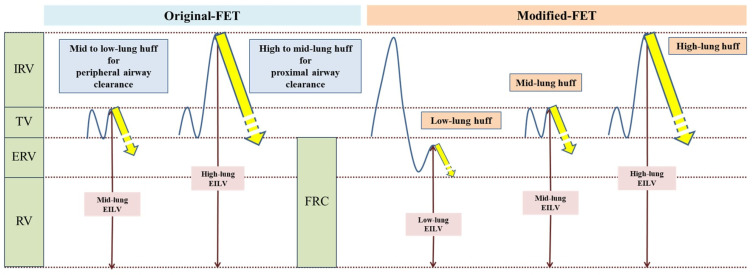
Schematic comparison of original FET and mFET.

**Figure 2 reports-07-00057-f002:**
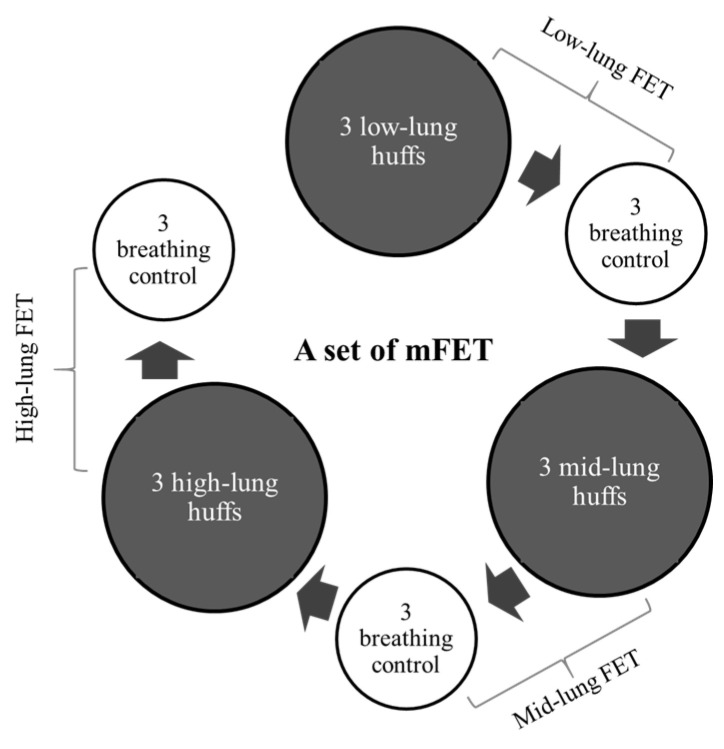
A set of modified mFET.

**Figure 3 reports-07-00057-f003:**
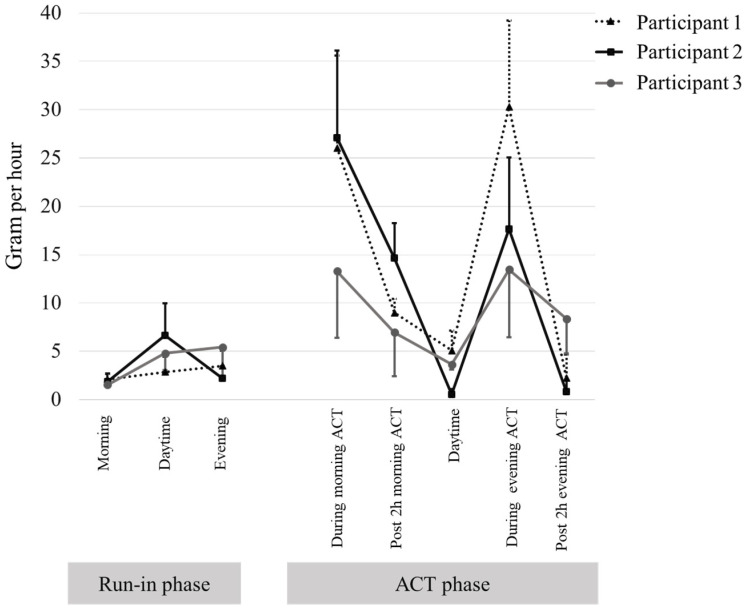
Rate per hour of sputum expectoration in the run-in phase and intervention phase. Data are presented as mean and standard deviation.

**Table 1 reports-07-00057-t001:** Participant selection criteria.

Inclusion Criteria	Exclusion Criteria	Withdrawal Criteria
Aged 18–75 years Confirmed diagnosis of bronchiectasis with or without other respiratory disease (e.g., COPD) by chest physician (X-rays, CT chest) Reported daily sputum production (>10 mL or 4 teaspoons) or median to coarse crackles or rhonchi during listening of lung auscultation and rhonchal fremitus Good cooperation, communication, and able to follow commands Stable vital signs (temperature, heart rate, blood pressure, respiratory rate)	History of pulmonary embolism, empyema, pleural disease, massive hemoptysis SpO_2_ < 90% at room air Active pulmonary tuberculosis Exacerbation within 3 months Any cancer Cardiac dysfunction and hemodynamic instability History of brain aneurism or retinal detachment, recent facial, oral, or skull surgery, or abdominal aneurism Neurological: old CVD and SCI Pregnancy	Hemoptysis Changing the medication during trial

**Table 2 reports-07-00057-t002:** Sputum collection during the run-in phase.

**Sputum Expectorated at Run-in Phase**	**Time**	**Conditions**	**Sputum Containers**
	6 a.m.–9 a.m.	Morning	C-1
	9 a.m.–6 p.m.	Daytime	C-2
	6 p.m.–9 p.m.	Evening	C-3

**Table 3 reports-07-00057-t003:** Sputum collection during the ACT period.

**Sputum Expectorated at ACT Phase**	**Conditions**	**Sputum Containers**
	During ACT-morning session	C-1
	2 h after morning session	C-2
	Daytime (immediately after post 2 h to before evening ACT session)	C-3
	During ACT evening session	C-4
	2 h after evening session	C-5

**Table 4 reports-07-00057-t004:** Demographic characteristics of the participants.

	Participant 1	Participant 2	Participant 3
Age	53	67	62
Gender	Male	Female	Female
Body mass index	15.92	19.72	22.21
Modified research council scale for dyspnea	3	1	1
Lung function			
IC, mean ± SD L	NA	1.36	1.34
IC, mean ± SD % predicted	NA	79.5	96.1
SVC, mean ± SD L	NA	1.46	1.73
SVC, mean ± SD % predicted	NA	61.3	82.3
FEV1, mean ± SD L	NA	1.05	1.38
FEV1, mean ± SD %predicted	NA	60	79
FVC, mean ± SD L	NA	1.25	1.58
FVC, mean ± SD % predicted	NA	58	76
FEV1/FVC, mean ± SD %	NA	83.91	87.22
PEF, mean ± SD L/min	NA	4.98	5.59
PEF, mean ± SD % predicted	NA	98	111
Comorbid disease	COPD	Hypertension	-
Medications			
Tiotropium bromide monohydrate + Olodaterol	✓	✓	✓
Salbutamol	✓	-	-
Vital sign at rest			
Heart rate, bpm	98	82	78
Respiratory rate, rpm	25	23	19
Peripheral oxygen saturation, %	95	97	97
Blood pressure, mm Hg	118/63	141/69	106/61
Daily sputum production, grams (from 6 a.m. to 9 p.m. of 3 days average)	42.47	72	63.98
Transportation to hospital			
Distance, km	52 km	134 km	151 km
Approximated driving duration	1 h 10 min	2 h	2 h 10 min

IC; inspiratory capacity, SVC; slow vital capacity, FVC; forced vital capacity, FEV1; forced expiratory volume in one second, PEF; peak expiratory flow, L; liter, NA; not available, ✓; current medication. The data of lung function for participant 1 was not available due to spontaneous cough during the spirometry procedure.

**Table 5 reports-07-00057-t005:** The CAT score of each participant.

Items	Pre-ACT Phase			Post-ACT Phase			Change Score		
	Pt. 1	Pt. 2	Pt. 3	Pt. 1	Pt. 2	Pt. 3	Pt. 1	Pt. 2	Pt. 3
1. Cough	4	4	3	3	3	2	−1	−1	−1
2. Phlegm in chest	5	3	3	3	4	1	−2	−1	−2
3. Chest tightness	0	0	0	0	0	0	0	0	0
4. Breathless on walk up a hill or one flight of stairs	3	0	3	0	0	1	−3	0	−2
5. Limitation of activities at home	2	0	0	0	0	0	−2	0	0
6. Confidence of leaving home	4	4	0	0	2	0	−4	−2	0
7. Sleep soundly	2	3	0	0	1	0	−2	−2	0
8. Energy	3	4	3	1	0	0	−2	−4	−3
Total CAT score	23	18	12	7	10	4	−16	−8	−8

## Data Availability

Dataset available on request from the authors.

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
