# Peer review of "The Effects of Hybrid Tele Airway Clearance in Bronchiectasis Patients: A Case Series"

_reports, 2024, doi:10.3390/reports7030057_

Round 1

Reviewer 1 Report

Comments and Suggestions for Authors

I believe that the study is limited by the number of patients, but it importantly highlights the use of a hybrid telemedicine approach to one of the most important components of treatment of patients with bronchiectasis.  I believe that this information is relevant and important. 

Comments:

1) please remove persistent and change to permanent 

2) please clarify if its 10cc or 30 cc of sputum production (table vs text)

3) table 4: please clarify what is HT (hypertension?)  

Author Response

1. Summary

Thank you very much for taking the time to review this manuscript. Please find the detailed responses below and corrections highlighted in the re-submitted revised manuscript.

2. Questions for General Evaluation

Reviewer’s Evaluation

Response and Revisions

Does the introduction provide sufficient background and include all relevant references?

Yes/Can be improved/Must be improved/Not applicable

[Please give your response if necessary. Or you can also give your corresponding response in the point-by-point response letter. The same as below]

Are all the cited references relevant to the research?

Yes/Can be improved/Must be improved/Not applicable

Is the research design appropriate?

Yes/Can be improved/Must be improved/Not applicable

Are the methods adequately described?

Yes/Can be improved/Must be improved/Not applicable

Are the results clearly presented?

Yes/Can be improved/Must be improved/Not applicable

Are the conclusions supported by the results?

Yes/Can be improved/Must be improved/Not applicable

3. Point-by-point response to Comments and Suggestions for Authors

Comments 1: Please remove persistent and change to permanent.

Response 1: Thank you for your comment. We agree with this comment. Therefore, we have removed "persistent" and changed it to "permanent" (in line 31 of revised manuscript).

Comments 2: Please clarify if its 10cc or 30 cc of sputum production (table vs text).

Response 2: Thank you for pointing this out. We have revised the text to indicate 10 cc (line 76 of revised manuscript).

Comments 3: Table 4: please clarify what is HT (hypertension?) 

Response 3: Thank you for pointing this out. We have revised HT to hypertension in table 4.

Reviewer 2 Report

Comments and Suggestions for Authors

Dear Editor,

I was pleased to review the manuscript entitled ''The effects of hybrid tele airway clearance in bronchiectasis patients: a case series''. Nwe et al. aimed to evaluate the preliminary effects of a hybrid tele-supervised airway clearance protocol on secretion clearance, health-related quality of life and patient satisfaction in patients with bronchiectasis. They conducted a single-arm experimental pilot case series with three participants, conducting six ACT sessions over three days, one on-site supervision and five tele-supervised sessions. The study showed increased expectoration rates and significant improvements in CAT scores during each ACT session.

Although the number of cases is small, it can be considered as a preliminary study to be evaluated with more patients in the future.

Sincerely

Comments on the Quality of English Language

Minor editing of English language required

Author Response

1. Summary

Thank you very much for taking the time to review this manuscript. Please find the detailed responses below and corrections highlighted in the re-submitted revised manuscript.

2. Questions for General Evaluation

Reviewer’s Evaluation

Response and Revisions

Does the introduction provide sufficient background and include all relevant references?

Yes/Can be improved/Must be improved/Not applicable

[Please give your response if necessary. Or you can also give your corresponding response in the point-by-point response letter. The same as below]

Are all the cited references relevant to the research?

Yes/Can be improved/Must be improved/Not applicable

Is the research design appropriate?

Yes/Can be improved/Must be improved/Not applicable

Are the methods adequately described?

Yes/Can be improved/Must be improved/Not applicable

Are the results clearly presented?

Yes/Can be improved/Must be improved/Not applicable

Are the conclusions supported by the results?

Yes/Can be improved/Must be improved/Not applicable

3. Point-by-point response to Comments and Suggestions for Authors

Comments 1: I was pleased to review the manuscript entitled ''The effects of hybrid tele airway clearance in bronchiectasis patients: a case series''. Nwe et al. aimed to evaluate the preliminary effects of a hybrid tele-supervised airway clearance protocol on secretion clearance, health-related quality of life and patient satisfaction in patients with bronchiectasis. They conducted a single-arm experimental pilot case series with three participants, conducting six ACT sessions over three days, one on-site supervision and five tele-supervised sessions. The study showed increased expectoration rates and significant improvements in CAT scores during each ACT session.

Although the number of cases is small, it can be considered as a preliminary study to be evaluated with more patients in the future.

Response 1: Thank you for your comment. We agree that the number of cases is small. Therefore, we recommended further research with a larger sample size in the discussion (line 301 -302 of revised manuscript).

  1. Response to Comments on the Quality of English Language

Point 1: Minor editing of English language required

Response 1: Thank you for pointing this out. We have carefully revised the grammar as suggested.

Reviewer 3 Report

Comments and Suggestions for Authors

The authors present an interesting study to evaluate the preliminary effects of a hybrid tele-supervised airway clearance protocol on secretion clearance, health-related quality of life, and patient satisfaction in bronchiectasis patients. 

Although the sample size is small, the study protocol is well-structured. I would recommend a larger sample size.

Can authors suggest how the data from this study can be extrapolated to global healthcare, especially in low-middle-income countries?

Include suggested future recommendation

Author Response

1. Summary

Thank you very much for taking the time to review this manuscript. Please find the detailed responses below and corrections highlighted in the re-submitted revised manuscript.

2. Questions for General Evaluation

Reviewer’s Evaluation

Response and Revisions

Does the introduction provide sufficient background and include all relevant references?

Yes/Can be improved/Must be improved/Not applicable

[Please give your response if necessary. Or you can also give your corresponding response in the point-by-point response letter. The same as below]

Are all the cited references relevant to the research?

Yes/Can be improved/Must be improved/Not applicable

Is the research design appropriate?

Yes/Can be improved/Must be improved/Not applicable

Are the methods adequately described?

Yes/Can be improved/Must be improved/Not applicable

Are the results clearly presented?

Yes/Can be improved/Must be improved/Not applicable

Are the conclusions supported by the results?

Yes/Can be improved/Must be improved/Not applicable

3. Point-by-point response to Comments and Suggestions for Authors

Comments 1: Although the sample size is small, the study protocol is well-structured. I would recommend a larger sample size.

Response 1: Thank you for your comment. We agree that the number of cases is small. Therefore, we recommended further research with a larger sample size in the discussion (line 30 -302 of revised manuscript).

Comments 2: Can authors suggest how the data from this study can be extrapolated to global healthcare, especially in low-middle-income countries?

Include suggested future recommendation

Response 2: Thank you for your suggestions. Accordingly, we have discussed the implementation (line 286-293 of revised manuscript) and future recommendation (line 301-307 of revised manuscript) in discussion.

Round 2

Reviewer 1 Report

Comments and Suggestions for Authors

No further comments.